# Numerical Study on Laser Shock Peening of Pure Al Correlating with Laser Shock Wave

**DOI:** 10.3390/ma15207051

**Published:** 2022-10-11

**Authors:** Mingxiao Wang, Cheng Wang, Xinrong Tao, Yuhao Zhou

**Affiliations:** 1Huadian Electric Power Research Institute Co., Ltd., National Energy Distributed Energy Technology R&D Center, Key Laboratory of Energy Storage and Building Energy-Saving Technology of Zhejiang Province, Hangzhou 310030, China; 2College of Electrical Engineering, Zhejiang University, Hangzhou 310013, China; 3School of Mechanical Engineering, Anhui University of Science and Technology, Huainan 232001, China

**Keywords:** laser shock processing, LSP-induced shock wave, compressive residual stress field, dislocation density, grain refinement, numerical simulation

## Abstract

Laser shock peening (LSP) is an innovative and promising surface strengthening technique of metallic materials. The LSP-induced plastic deformation, the compressive residual stresses and the microstructure evolution are essentially attributed to the laser plasma-induced shock wave. A three-dimensional finite element model in conjunction with the dislocation density-based constitutive model was developed to simulate the LSP of pure Al correlating with the LSP-induced shock wave, and the predicted in-depth residual stresses are in reasonable agreement with the experiment results. The LSP-induced shock wave associated with the laser spot diameter of 8.0 mm propagates in the form of the plane wave, and attenuates exponentially. At the same time, the propagation and attenuation of the LSP-induced shock wave associated with the laser spot diameter of 0.8 mm are in the form of the spherical wave. The reflection of the LSP-induced shock wave at the bottom surface of the target model increases the plastic deformation of the target bottom, resulting in the increase of dislocation density and the decrease of dislocation cell size accordingly. Reducing the target thickness can significantly increase the reflection times of the LSP-induced shock wave at the bottom and top surfaces of the target model, which is considered to be conductive to the generation of the compressive residual stress field and grain refinement.

## 1. Introduction

Laser shock peening (LSP) is one of the most promising techniques for surface enhancement, and is widely used for improving the wearing quality, corrosion resistance and fatigue performance by inducing the beneficial compressive residual stress field and grain refinement in the processed materials [1,2,3,4,5]. In comparison with the conventional shot peening (SP) and ultrasonic shot peening (USP) [6,7,8], LSP not only has good operating controllability and geometric adaptability, but also can produce a deeper compressive residual stress field and the better preservation of surface roughness [9].

Different from the SP or USP, LSP is a contact-free surface treatment technique. The laser beam with the ultra-high power density ($\sim \mathrm{GW}/{\mathrm{cm}}^{2}$) and ultra-short laser pulse (~ns) passes through the transparent confining overlay (usually water or glass) and irradiates the ablative layer covering the material surface to be peened. The excessive heat induced by the laser irradiation vaporizes and ionizes the ablative layer that then turns into the plasma. The plasma expands rapidly by continuing to absorb the laser radiation, and the plasma-induced shock wave with the ultra-high peak pressure is resultantly formed. Owing to the constraint effect of the confining layer, the pressure of the laser plasma-induced shock wave is amplified. As the laser shock wave propagates into the material target, when the peak pressure of the LSP-induced stress wave exceeds the dynamic yield strength of the metallic materials, the plastic deformation with ultra-high strain rate ($10^{6}\;s^{-1}$) occurs, resulting in the generation of the compressive residual stress field, dislocation arrays, grain refinement and phase changes [10,11,12].

In recent years, a great deal of studies on the LSP have been reported, and most of them mainly focus on the effects of the LSP parameters (laser power density, spot size, laser pulse, overlap rate of the laser spot, etc.) on the results of LSP, such as the compressive residual stresses, surface roughness and microstructure evolutions [13,14,15]. Considering that the laser plasma-induced shock wave is responsible for the surface strengthening effect of LSP, the LSP-induced shock wave can be experimentally detected by several devices, such as a velocimetry interferometer system for any reflector (VISAR) interferometer device and the piezoelectric thin-film polyvinylidene fluoride (PVDF). Zhou et al. [16] used an optically transparent target and interference imaging technology for the in situ observation of the laser-induced plasma in the air and corresponding stress waves in the target, and found that the temporal profiles of laser pulse have a significant impact on the fluid dynamics and ionization characteristics of the laser-induced plasma, which in turn affects the stress distribution of target material.

Although it is feasible to carry out the in situ exploration of the propagation and attenuation of the LSP-induced shock wave in the process of LSP of metals by utilizing the available experiment devices [17], the finite element method has been increasingly favored and frequently used by the researchers with the development of computer technology and numerical calculation [18,19,20,21]. Jiang et al. [22] developed a three-dimensional semi-infinite finite element model to investigate the generation and propagation of the laser shock wave in Al2024 generated by the laser-induced plasma pressure, and the prediction results were verified by the PVDF measurements. Nie et al. [23] developed two finite element models with thicknesses of 5 mm and 1 mm, and the bottom of the 5 mm-thickness target model was added the infinite elements to eliminate the affection of the laser plasma-induced shock wave reflection. Compared with the infinite thickness model, the propagation law of the LSP-induced shock wave and the dynamic response of material were analyzed, and the simulation results show that the reflection of the LSP-induced shock wave has an important impact on the LSP-induced residual stresses in the thin (sheet) model. Yang et al. [24] explored the formation mechanism of the surface residual stress hole induced by LSP by analyzing the propagation and reflection of the laser plasma-induced shock wave, and found that the residual stress hole occurring in the thin plate subjected to the two-side laser shock peening is attributed to the co-action between the convergence of unloading surface stress emitted from the boundary of laser spot and axial stress wave going back and forth between two shocked side surfaces. Zhang et al. [25] carried out the three-dimensional finite element simulation of two side laser shock processing of the 7075-T7351 alloy plate with the thickness of 2 mm, and the dynamic propagation and attenuation process of the LSP-induced shock waves in the thickness direction of the plate was investigated, the simulation results reveal that the interaction between target material and LSP-induced stress wave gradually decays with the increase in the propagation distance. Xiang et al. [26] utilized the finite element method to investigate the dynamic propagation characteristics of stress waves induced by LSP with the round and square spots, and found that the dynamic stress distributions induced by the two kinds of spots are quite different at the same time, but the final residual stress distributions are basically the same. Additionally, Xiang et al. [27] also preformed the double-sided laser shot peening of the thin 2024-T351 alloy blade through the experiments and finite element simulations, and further investigated the mechanism of the laser plasma-induced shock wave reflection-coupling on the LSP-induced residual stress distribution, the obtained results show that the alternate double-sided laser shot peening model could produce a better residual stress distribution. 

Although a large number of researches have been carried out and attempted to reveal the formation mechanism of the LSP-induced compressive residual stress field by using the finite element method, whereas there are rarely reports on the microstructure evolutions of the target material during the LSP process, and the effects of the spatial distribution of the laser plasma-induced shock wave pressure on the dynamic propagation and attenuation of the LSP-induced shock wave have been little studied. Therefore, this paper is devoted to numerical simulations of the LSP process correlating with the laser shock wave resulting from the loading of the different spatially-distributed laser plasma-induced shock wave pressures, and the effects of the dynamic propagation, attenuation and reflection of the LSP-induced shock wave on the resultant plastic deformation, compressive residual stress field, dislocation density evolution and grain refinement are further investigated in detail. For simplicity, it should be noted that the finite element modeling of LSP in this work has not taken into account the influences of the actual anisotropy of the material grain-level response on the numerical simulations. 

## 2. Finite Element Modeling of LSP

### 2.1. Dislocation Density-Based Constitutive Model

In order to investigate the LSP-induced microstructure evolutions in terms of the dislocation density evolution and grain refinement, the dislocation density-based constitutive model was employed to characterize the dynamic mechanical behavior of the target material in the LSP process. The dislocation density-based constitutive model was proposed by Estrin et al. [28,29,30], and the dynamic flow stress in the constitutive model can be expressed as [31]
(1)$$\sigma_{y}=\sigma_{1}+M\alpha Gb\left[f\sqrt{\rho_{w}}+\left(1-f\right)\sqrt{\rho_{c}}\right]\cdot {\left(\frac{M\dot{\bar{\varepsilon }}}{{\dot{\gamma }}_{0}}\right)}^{1/m^{*}}$$
where, $\sigma_{1}$ is the strain-independent contribution originating from the resistance to dislocation glide, *M* is Taylor factor and M=3.06 in general, α is a constant and α=0.25, *G* is the shear modulus, *b* is the magnitude of Burgers vector ($b=2.86\times 10^{-7}\;\mathrm{mm}$), *f* is the volume fraction of the dislocation cell walls and is related to the equivalent plastic strain ($\bar{\varepsilon }$), the evolution of *f* can be described as
(2)$$f=f_{\infty }+\left(f_{0}-f_{\infty }\right)\exp\left(-\frac{M\bar{\varepsilon }}{{\tilde{\gamma }}^{r}}\right)$$
where $f_{\infty }$ and $f_{0}$ are the saturation and initial values of *f*, respectively, ${\tilde{\gamma }}^{r}$ is a constant with the value of 3.2 quantifying the descent of *f*, ${\dot{\gamma }}_{0}$ is the reference shear strain rate, $m^{*}$ is the stain rate sensitivity parameter, $\rho_{w}$ and $\rho_{c}$ are the dislocation densities within cell walls and cell interiors, respectively. The evolutions of $\rho_{w}$ and $\rho_{c}$ are dependent on the equivalent plastic strain rate ($\dot{\bar{\varepsilon }}$) and are expressed as
(3)$$\left\{ \begin{array}{l} \frac{{\dot{\rho }}_{c}}{M\dot{\bar{\varepsilon }}}=\frac{\alpha^{*}\sqrt{\rho_{w}}}{\sqrt{3}b}-\frac{6\beta^{*}}{bd{\left(1-f\right)}^{1/3}}-k^{*}\rho_{c}{\left(\frac{M\dot{\bar{\varepsilon }}}{{\dot{\gamma }}_{0}}\right)}^{-1/n^{*}}\\ \frac{{\dot{\rho }}_{w}}{M\dot{\bar{\varepsilon }}}=\frac{\sqrt{3}\beta^{*}\left(1-f\right)\sqrt{\rho_{w}}}{fb}+\frac{6\beta^{*}{\left(1-f\right)}^{2/3}}{bdf}-k^{*}\rho_{w}{\left(\frac{M\dot{\bar{\varepsilon }}}{{\dot{\gamma }}_{0}}\right)}^{-1/n^{*}} \end{array}\right.$$
where, $\alpha^{*}$, $\beta^{*}$ and $k^{*}$ are the three material parameters and are respectively related to the nucleation, interaction and annihilation of dislocation, $n^{*}$ is a constant with the assumption of the isothermal treatment processes, *d* represents the averaged dislocation cell size and is inversely proportional to the square root of the total dislocation density, i.e.,
(4)$$d=K/\sqrt{\rho_{t}}$$
where *K* is a constant and K=30 [32], $\rho_{t}$ is the total dislocation density and is calculated by
(5)$$\rho_{t}=f\rho_{w}+\left(1-f\right)\rho_{c}$$

By developing a user subroutine program of VUMAT, the dislocation density-based constitutive model consisting of Equations (1)–(5) was implemented into the finite element codes of ABAQUS/Explicit for carrying out the finite element simulation of LSP of pure Al. The relative material parameters of the dislocation density-based constitutive model with respect to the pure Al are listed in Table 1 [32]. It is noted that the dislocation density-based constitutive model was proposed with the assumption that the metallic materials are continuous, homogeneous and isotropic. The micro voids in the real metallic materials are not taken into consideration, which would have an impact on the finite element computation results.

### 2.2. Laser Shock Wave Pressure Model

The strengthening effects of LSP are dependent on the laser shock wave pressure, which is generated by the expanding plasma resulting from the ionization of the laser-irradiated ablative layer. According to the research of Fabbro et al. [33], the peak value of the laser plasma-induced shock wave pressure profile is correlated with the laser power density and the reduced shock impedance, i.e.,
(6)$$P_{\max}=0.01\sqrt{\frac{\alpha }{2\alpha +3}}\cdot \sqrt{\frac{2Z_{1}Z_{2}}{Z_{1}+Z_{2}}}\cdot \sqrt{I_{p}}$$
where, $P_{\max}$ is the peak pressure, α is a correction factor representing the transformed ratio of laser irritation energy, $I_{p}$ is the laser power density, $Z_{1}$ and $Z_{2}$ are the shock impedances of target material and confining medium, respectively.

The temporal distribution of the laser plasma-induced shock wave pressure profile is related to the laser pulse. The full width at half maximum (FWHM) of the laser shock wave pressure profile is assumed to be 2~3 times longer than that of laser pulse, and the FWHMs of the laser pulse and the corresponding laser shock wave pressure are 10 ns and 25 ns in the current study, respectively [34]. The temporal distribution of the normalized laser shock wave pressure can be characterized by a piecewise continuous function of time [35].
(7)$$\bar{P}\left(t\right)=\left\{ \begin{array}{l} \left(\tau_{P}-t\right)\cdot t/\tau_{P}^{2}\;\;\;\;\;\;0\leq t\leq \tau_{P}\\ \exp\left(\ln0.5\times \sqrt{\frac{t/\tau_{p}-1}{1-\sqrt{2}/2}}\right)\;\;\;t>\tau_{P} \end{array}\right.$$
where, *t* is the time, $\tau_{P}$ is the FWHM of the laser plasma-induced shock wave pressure profile.

The spatial distribution of the laser plasma-induced shock wave pressure profile is related to the laser spot size. For the smaller laser spot size, the laser plasma-induced shock wave pressure follows the Gaussian spatial distribution [36,37], whereas the laser shock wave pressure is considered to be distributed uniformly within the scope of the larger laser spot neglecting any boundary effects [38]. Therefore, the spatial distribution of the normalized laser shock wave pressure profile can be expressed as
(8)$$\bar{P}\left(r\right)=\left\{ \begin{array}{l} \exp\left(-2\frac{r^{2}}{D^{2}}\right)\;\;\mathrm{for}\;\mathrm{the}\;\mathrm{smaller}\;\mathrm{laser}\;\mathrm{spot}\;\mathrm{size}\\ 1\;\;\;\;\;\;\;\;\mathrm{for}\;\mathrm{the}\;\mathrm{smaller}\;\mathrm{laser}\;\mathrm{spot}\;\mathrm{size} \end{array}\right.$$
where, *r* is the radial distance from the laser spot center, and *D* is the laser spot diameter.

Combining the Equations (6)–(8), the temporal-spatial distribution of the laser plasma-induced shock wave profile is thereby described as
(9)$$P\left(r,t\right)=P_{\max}\cdot \bar{P}\left(t\right)\cdot \bar{P}\left(r\right)$$

The laser plasma-induced shock wave pressure expressed by Equation (9) was implemented into the ABAQUS/Explicit code by developing a user subroutine VDLOAD, and was imposed onto the top surface of the target model for the finite element simulation of LSP of pure Al. 

### 2.3. Finite Element Modelling of LSP

Considering the symmetry of the laser plasma-induced shock wave pressure, a 1/4 symmetric model of pure Al sample with the dimension of 20 mm×20 mm×20 mm was created. For improving the finite element computation efficiency, the infinite elements (CIN3D8) were used in the rest two sides instead of the finite elements for eliminating the reflection of LSP-induced stress wave on the sides, as shown in Figure 1. The three-dimensional eight-nodes linear brick elements with reduced integration and hourglass control (C3D8R in ABAQUS/Explicit) were used to mesh the target model of pure Al. Based on the sensitivity analysis of element size, the finest element size of 10 μm was adopted for the accurate simulation of the LSP-induced residual stresses and microstructure evolutions. The dislocation density-based constitutive model was assigned to the region meshed by C3D8R elements, and the linear elastic constitutive model was assigned to the CIN3D8 elements-meshed region. In order to investigate the effect of the target thickness on the dynamic propagation and attenuation of the shock wave, the residual stress field and microstructure evolution induced by LSP, three target models of pure Al sample with the thicknesses of 1 mm, 5 mm and 20 mm were established, respectively.

The top surface of the target model is exposed to the laser plasma-induced shock wave pressure. The peak pressures of the laser plasma-induced shock wave used are 0.6 GPa and 2.0 GPa, the diameters of the laser spot are 0.8 mm and 8.0 mm, respectively [34]. With respect to the same peak pressure and the temporal distribution of the laser plasma-induced shock wave pressure profile, both the uniform distribution and Gaussian distribution are employed as the spatial distribution of the laser plasma-induced shock wave pressure for exploring the effect of the spatial distribution of the laser plasma-induced shock wave pressure on the results of LSP of pure Al. The bottom surface is fixed in the z-axial direction, and the symmetric constraints are applied on the two symmetric surfaces, respectively, as shown in Figure 1.

Two successive explicit analysis steps are set up to simulate the process of LSP of pure Al. The first step associated with the analysis time 2 μs is utilized for applying the laser plasma-induced shock wave pressure, and the maximum time increment is smaller than 1 ns [39]. The second step with the analysis time of 20 μs is used for the spring-back computation of the elastic deformation. In the second analysis step, the laser plasma-induced shock wave pressure is removed, and a larger time increment is adopted for improving the finite element computation efficiency as much as possible. 

## 3. Finite Element Simulation Results

In order to verify the finite element computation time that is long enough to obtain the stable simulation results of LSP of pure Al, the evolutions of the internal energies and kinetic energies stored in the target material during the LSP process were outputted from the finite element simulation results of LSP, as shown in Figure 2. Once the laser plasma-induced shock wave pressure is imposed onto the top surface of target model, both the internal energies and kinetic energies stored in the target material instantaneously increase to the peak value, and then present the rapid-to-slow declining phase with the dynamic propagation and attenuation of the LSP-induced shock wave. As seen from Figure 2, within the analysis time of 5 μs, both the internal energies and kinetic energies resulting from the loading of the uniformly-distributed laser plasma-induced shock wave pressure are significantly larger than that resulting from the laser plasma-induced shock wave pressure with the Gaussian distribution. After the analysis time exceeds 10 μs, the internal energies reduce to stable values, and the residual kinetic energies also damp down to an insignificant level, implying that the finite element computation time of 20 μs in the second step is long enough to complete the spring-back analysis of elastic deformation. 

Taking the center of the laser spot as the starting point, a path along the thickness direction of the target model was established and named as ‘Path-Z’, as shown in Figure 1. Figure 3 compares the prediction results of LSP-induced in-depth residual stresses along the Path-Z with the corresponding experimental data in the two cases of $P_{\max}=0.6\;\mathrm{GPa}\;\&\;D=0.8\;\mathrm{mm}$ and $P_{\max}=2.0\;\mathrm{GPa}\;\&\;D=8.0\;\mathrm{mm}$. Due to the axis symmetry of the laser plasma-induced shock wave pressure, the in-plane stress components of $\sigma_{x}$ and $\sigma_{y}$ with respect to the same thickness are identical, therefore, the stress component of $\sigma_{x}$ is used to compare with the experimental data. It is concluded from Figure 3 that the finite element predictions of the in-depth residual stresses induced by LSP are in reasonable agreement with the experimental results [34], especially for the predictions of the LSP-induced compressive residual stress depths. Careful observation can be found that the prediction results of the in-depth residual stresses induced by LSP associated with the loading of the Gaussian distribution of laser plasma-induced shock wave pressure are more close to the experimental data in the case of $P_{\max}=0.6\;\mathrm{GPa}\;\&\;D=0.8\;\mathrm{mm}$, whereas the predicted in-depth residual stresses resulting from the LSP by applying the uniformly-distributed laser plasma-induced shock wave pressure agree better with the experimental data in the case of $P_{\max}=2.0\;\mathrm{GPa}\;\&\;D=8.0\;\mathrm{mm}$, revealing that the spatial distribution of the laser plasma-induced shock wave pressure gradually transforms to the uniform distribution from the Gaussian distribution with the increase of the laser spot size.

The LSP-induced residual stresses are the results of the recovery of the elastic deformations surrounding the plastically-deformed zone, and are essentially dependent on the plastic deformations caused by LSP. Figure 4 shows the in-depth equivalent plastic strains along the Path-Z, which can be used for describing the plastic deformations quantitatively. Under the same LSP conditions (the peak pressure of laser plasma-induced shock wave and laser spot size), with regard to the same depth, the equivalent plastic strain induced by LSP associated with the loading of the uniformly-distributed laser plasma-induced shock wave pressure is larger than that associated with the loading of the Gaussian distribution of laser plasma-induced shock wave pressure, which is consistent with the depth of the LSP-induced compressive residual stress field. 

The in-depth equivalent plastic strain induced by LSP is essentially related to the dynamic propagation and attenuation of the LSP-induced shock wave during the LSP process. When the LSP-induced shock wave pressure is larger than the dynamic yield stress of the target material, the plastic deformation of the target material is produced resultantly. Figure 5 presents the dynamic propagation and attenuation of the LSP-induced shock wave in the two cases of $P_{\max}=0.6\;\mathrm{GPa}\;\&\;D=0.8\;\mathrm{mm}$ and $P_{\max}=2.0\;\mathrm{GPa}\;\&\;D=8.0\;\mathrm{mm}$. As seen in Figure 5a1–a8,b1–b8, in the case of $P_{\max}=2.0\;\mathrm{GPa}\;\&\;D=8.0\;\mathrm{mm}$, the LSP-induced shock wave propagates and attenuates along the target thickness direction in the form of the plane wave, after the LSP-induced shock wave propagation and attenuation of 3 μs, the LSP-induced shock wave approaches the bottom surface of the target model with the thickness of 20 mm. Careful observation of Figure 5a1–a8,b1–b8 can find that, with respect to the same time, the peak pressure of the LSP-induced shock wave resulting from the loading of the uniformly-distributed laser plasma-induced shock wave pressure is larger than that resulting from the loading of the laser plasma-induced shock wave pressure with the Gaussian distribution, which is attributed to that the more energy of laser beam is imported into the target material by applying the uniformly-distributed laser plasma-induced shock wave pressure. Different from the case of $P_{\max}=2.0\;\mathrm{GPa}\;\&\;D=8.0\;\mathrm{mm}$, in the case of $P_{\max}=0.6\;\mathrm{GPa}\;\&\;D=0.8\;\mathrm{mm}$, as seen in Figure 5c1–c8,d1–d8, since the laser spot size (D=0.8 mm) is much smaller than that in the case of $P_{\max}=2.0\;\mathrm{GPa}\;\&\;D=8.0\;\mathrm{mm}$, the LSP-induced shock wave rapidly transforms into the spherical wave from the plane wave for the propagation and attenuation. By comparing Figure 5c1–c8,d1–d8, the more significant LSP-induced shock wave propagation in the form of the spherical wave can be observed by applying the laser plasma-induced shock wave pressure with the Gaussian distribution, and whether the uniform distribution or Gaussian distribution of the laser plasma-induced shock wave pressure, after the LSP-induced shock wave propagation of 1 μs, the peak pressure of the LSP-induced shock wave reduces to 20 MPa that is much smaller than the dynamic yield stress of the target material.

Figure 6 shows the evolution of the peak pressure of the LSP-induced shock wave propagating along Path-Z. In the case of $P_{\max}=2.0\;\mathrm{GPa}\;\&\;D=8.0\;\mathrm{mm}$, as seen in Figure 6a, the peak pressure of the LSP-induced shock wave decreases exponentially with the depth of the target model, and the double exponential function is used to fitting the decrease of the peak values of the LSP-induced shock wave pressure. According to the two double exponential functions in Figure 6, the decay rate of the LSP-induced shock wave pressure associated with the loading of the Gaussian distribution of the laser plasma-induced shock wave pressure is larger than that associated with the loading of the uniform distribution of the laser plasma-induced shock wave pressure. On the other hand, in the case of $P_{\max}=0.6\;\mathrm{GPa}\;\&\;D=0.8\;\mathrm{mm}$, as seen in Figure 6b, when compared with the case of $P_{\max}=2.0\;\mathrm{GPa}\;\&\;D=8.0\;\mathrm{mm}$, the decay rate of the LSP-induced shock wave pressure associated with the loading of the uniformly-distributed laser plasma-induced shock wave pressure is also smaller than that associated with the loading of the Gaussian distribution of the laser plasma-induced shock wave pressure, whereas the decay rate of the LSP-induced shock wave pressure is much larger due to the much smaller laser spot size resulting in the attenuation of the LSP-induced shock wave in the form of the spherical wave. According to the research of Fan et al. [40], the decay rate of the spherical wave is larger than that of the plane wave.

As a result of the dynamic propagation and attenuation of the LSP-induced shock wave, the plastic deformations caused by LSP not only account for the beneficial compressive residual stress field, but also is responsible for the increase of dislocation density and decrease of the dislocation cell size, which can be described by the dislocation density-based constitutive model consisting of Equations (1) - (5). Figure 7 shows the in-depth dislocation densities and cell sizes of the LSPed target model, which are consistent with the in-depth equivalent plastic strains distributed along Path-Z. Under the laser plasma-induced shock wave pressure, whether the maximum dislocation or the minimum cell size is located on the LSPed surface, and both the increment of dislocation density and the decrement of the cell size decrease with the increase of the depth. With respect to the same depth, in comparison with the results of the LSP with the loading of the Gaussian distribution of laser plasma-induced shock wave pressure, the increased dislocation density resulting from the LSP with the loading of the uniformly-distributed laser plasma-induced shock wave pressure is larger, and the decreased cell size is smaller correspondingly. Additionally, as seen from Figure 7, when compared to the increased dislocation density and the decreased cell size induced by LSP in the case of $P_{\max}=2.0\;\mathrm{GPa}\;\&\;D=8.0\;\mathrm{mm}$, the increased dislocation density and the decreased cell size induced by LSP in the case of $P_{\max}=0.6\;\mathrm{GPa}\;\&\;D=0.8\;\mathrm{mm}$ are much insignificant.

## 4. Effects of Target Thicknesses

In order to further investigate the effect of target thicknesses on the results of LSP in terms of the residual stresses, dislocation densities and cell sizes, two target models of pure Al with thicknesses of 1 mm and 5 mm were established, respectively. Different from the finite element simulation results of the 20 mm-thickness target model, the LSP-induced shock wave would reflect at the bottom surface of the target models with the thicknesses of 1 mm and 5 mm, and the reflected wave of the LSP-induced shock wave would have an impact on the results of LSP. Based on the study of the effect of the spatial distribution of laser plasma-induced shock wave pressure imposing onto the target top surface on the results of LSP, the uniformly-distributed laser plasma-induced shock wave pressure associated with the peak pressure of 2.0 GPa and laser spot diameter of 8.0 mm ($P_{\max}=2.0\;\mathrm{GPa}\;\&\;D=8.0\;\mathrm{mm}$) was imposed onto the top surfaces of the target models with the thicknesses of 1 mm and 5 mm.

Figure 8 presents the dynamic propagation and attenuation of the LSP-induced shock wave in the target thickness direction. As seen in Figure 8a1–a8, in the case of LSP of the 5 mm-thickness target model, after the LSP-induced shock wave propagates in the form of the plane wave for 800 ns, the LSP-induced shock wave approaches the bottom surface of target model. Owing to the reflection of the LSP-induced shock wave at the target bottom surface, the peak pressure of the LSP-induced shock wave reaches 558 MPa at the time of 961 ns. Subsequently, the reflected wave of the LSP-induced shock wave inversely propagates toward the top surface of the target model with the decreasing peak pressure of the shock wave due to the increasing attenuation of the LSP-induced shock wave. The more significant reflection of the LSP-induced shock wave at the target bottom surface can be seen in Figure 8b1–b8. Different from Figure 8a1–a8, in the case of LSP of the 1 mm-thickness target model, the first reflection of the LSP-induced shock wave occurs at the time of 224 ns, resulting in the peak pressure of the shock wave reaching 1.67 GPa. 

Figure 9 presents the evolution of the peak pressure of the LSP-induced shock wave propagating along the thickness direction, which accords well with the dynamic propagation, attenuation and reflection of the LSP-induced shock wave in Figure 8. Due to the reflection of the LSP-induced shock wave and the interaction between the reflected wave and the LSP-induced shock wave, the attenuation law of the shock wave in Figure 9 is different from that in Figure 6 which presents the exponential decay law of LSP-induced laser shock wave.

Figure 10 shows the evolutions of the LSP-induced shock wave pressures of the elements locating in Path-Z with the different distances from the LSPed surface (target top surface). As seen from Figure 10a, in the case of the LSP of the 5 mm-thickness target model, a significant increase of the peak pressure of the LSP-induced shock wave at the target bottom surface can be observed. In addition, careful observation can further find that the peak pressures of the LSP-induced shock wave of the elements closing to the target bottom surface become increasingly larger with the decrease of the distance from the target bottom surface, which is attributed to the superposition of the reflected wave and the LSP-induced shock wave. Figure 10b1,b2 shows the evolutions of the LSP-induced shock wave pressures in the case of LSP of the 1 mm-thickness target model. It can be seen from Figure 10b1 that there are three reflections of the LSP-induced shock wave at the target bottom surface and two reflections of the LSP-induced shock wave at the target top surface within the computation time of 2 μs. Due to the increasing attenuation of the LSP-induced shock wave during its propagation process, the peak pressures of the reflected waves gradually decrease with the increases of the reflection times. The first reflection of the LSP-induced shock wave at the bottom surface of the 1mm-thickness target model is presented separately in Figure 10b2, it is evident that the reflected wave has made a big difference to the propagation and attenuation of the shock wave along the Path-Z, whereas only the evolution of the LSP-induced shock wave of the element in the LSPed surface is insignificantly influenced by the reflected wave. 

With the LSP-induced shock wave and the reflected wave, the plastic deformations of the LSPed target material are produced resultantly. Figure 11 shows the evolutions of the equivalent plastic strains of the elements located in the top and bottom surfaces of the target model during the LSP process. For the LSP of the pure Al target model with the thicknesses of 1 mm and 5 mm, in the case of $P_{\max}=2.0\;\mathrm{GPa}\;\&\;D=8.0\;\mathrm{mm}$, there is little difference in the evolutions of the equivalent plastic strains of the elements in the top surface (the LSPed surface), implying that the reflected wave of the LSP-induced shock wave has little impact on the plastic deformation of the LSPed surfaces with the target thickness larger than 1mm. However, as seen in Figure 11, owing to the superposition of the LSP-induced shock wave and its reflected wave, the equivalent plastic strain of the target bottom surface increases. In the 5 mm-thickness target model, the equivalent plastic strain at the target bottom increases only once, since the significant reflection of the LSP-induced shock wave occurs for only once. Unexpectedly, it is interesting to find that the equivalent plastic strain at the bottom surface of the 1 mm-thickness target increases by five times correlating with the reflection times of the LSP-induced shock wave, and the increased equivalent plastic strain even becomes larger than that in the LSPed surface.

Figure 12 compares the in-depth equivalent plastic strains resulting from the LSP of the target models with thicknesses of 1 mm, 5 mm and 20 mm. It is evident that the reduction of the target thickness is conductive to producing the plastic deformation of the target bottom surface, especially for the smaller thickness (~1 mm), the plastic deformation of the target bottom surface caused by the reflected wave pressure even becomes larger than that of the LSPed surface.

Depending on the plastic deformations caused by the LSP-induced shock wave and its reflected wave, the in-depth residual stresses resulting from the recovery of the elastic deformations surrounding the plastically-deformed zone are shown in Figure 13. For the 1 mm-thickness target model of pure Al, the compressive residual stresses closing to the target bottom surface are larger than that closing to the LSPed surface. However, the distribution of the in-depth residual stresses resulting from the finite element simulation of LSP of the 5 mm-thickness target model is much similar to the results of LSP of the 20 mm-thickness target model, although the maximum compressive residual stress is closing to the target bottom surface, which is different from that of the 20 mm-thickness target model. The reason could be explained that the smaller plastic deformations at the bottom of the 5 mm-thickness target model, as shown in Figure 11, may have not made a big difference in the in-depth residual stresses induced by LSP. 

By utilizing the dislocation density-based constitutive model, both the increased dislocation density and the decreased cell size distributed along the Path-Z are presented in Figure 14. Correlating with the in-depth equivalent plastic strains, the significant increase in the dislocation density and the decrease in the cell size can be seen in Figure 14. When compared to the in-depth dislocation density and cell size resulting from the LSP of the target models with the thicknesses of 5 mm and 20 mm, in the case of LSP of the 1mm-thickness target model, owing to the multiple reflections of the LSP-induced shock wave, the dislocation density increases with the increase of the depth, and the cell size decreases with the depth correspondingly, and both the maximum dislocation density and the minimum cell size are located in the bottom surface of the target model. It thereby reveals that the reduction of the target thickness could increase the reflection times of LSP-induced shock wave, which would be conductive to the grain refinement induced by LSP. 

## 5. Conclusions

The three-dimensional finite element models coupling with the infinite elements were developed to simulate the LSP of pure Al, and the dislocation density-based constitutive model was employed to characterize the dynamic mechanical behavior of the LSPed target model. The effect of the dynamic propagation, attenuation and reflection of the LSP-induced shock wave on the resultant plastic deformation, compressive residual stress field and microstructure evolution were investigated in detail, and the obtained conclusions are drawn as follows.

For the larger laser spot size (D=8.0 mm), by imposing the uniformly-distributed laser plasma-induced shock wave pressure onto the target top surface, the predicted in-depth residual stresses are close to the experimental results; and the predictions outputting from the finite element model with the loading of the Gaussian distribution of the laser plasma-induced shock wave pressure are in good agreement with the experiment data for the smaller laser spot size (D=0.8 mm).The decay rate of the LSP-induced shock wave by applying the uniformly-distributed laser plasma-induced shock wave pressure is smaller than that associated with the laser plasma-induced shock wave pressure with the Gaussian distribution.The LSP-induced shock wave propagates in the form of the plane wave and attenuates exponentially in the case of $P_{\max}=2.0\;\mathrm{GPa}\;\&\;D=8.0\;\mathrm{mm}$, whereas the LSP-induced shock wave in the case of $P_{\max}=0.6\;\mathrm{GPa}\;\&\;D=0.8\;\mathrm{mm}$ propagates and attenuates in the form of the spherical wave.The reflection of the LSP-induced shock wave increases the shock wave pressure, resulting in the increase of the plastic deformation and dislocation density, and the decrease of the dislocation cell size.Reducing the target thicknesses would increase the reflection times of the LSP-induced shock wave, and is conductive to the compressive residual stress field and grain refinement induced by LSP.

## Figures and Tables

**Figure 1 materials-15-07051-f001:**
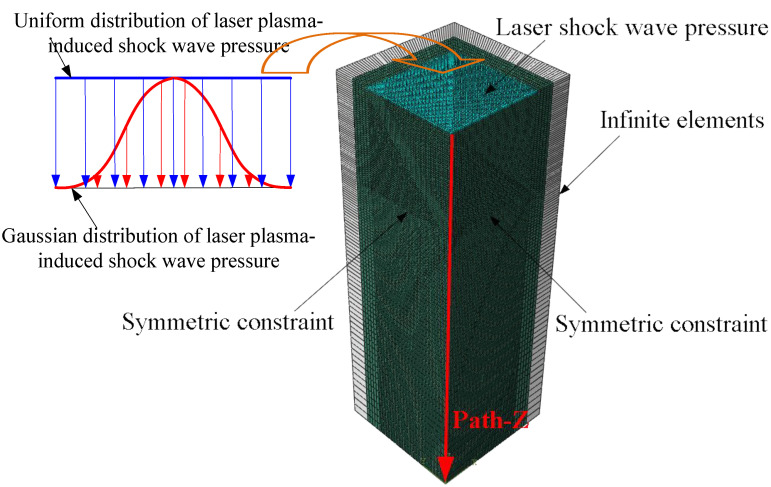
Finite element model of LSP of pure Al.

**Figure 2 materials-15-07051-f002:**
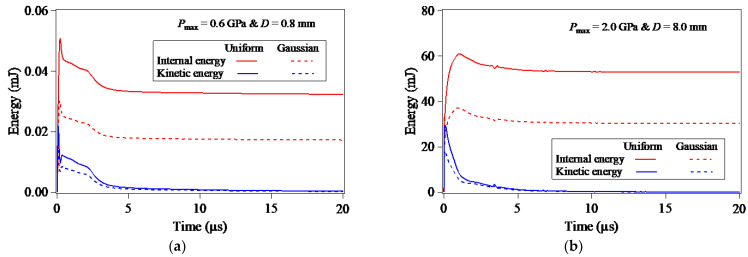
Evolutions of internal energies and kinetic energies stored in the target material during the LSP process: (**a**) $P_{\max}=0.6\;\mathrm{GPa}\;\&\;D=0.8\;\mathrm{mm}$; (**b**) $P_{\max}=2.0\;\mathrm{GPa}\;\&\;D=8.0\;\mathrm{mm}$.

**Figure 3 materials-15-07051-f003:**
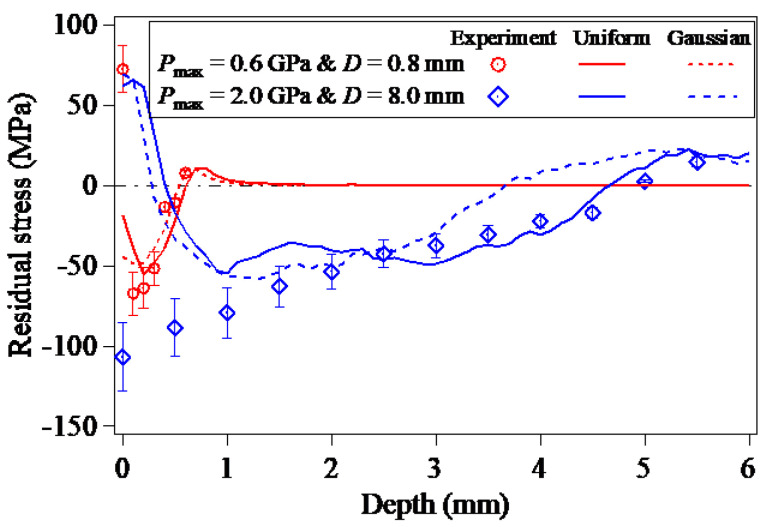
In-depth residual stresses induced by LSP [34].

**Figure 4 materials-15-07051-f004:**
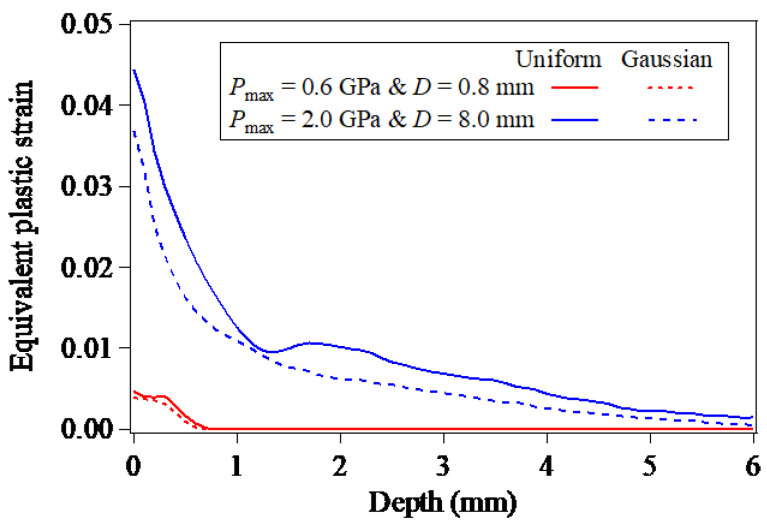
In-depth equivalent plastic strains induced by LSP.

**Figure 5 materials-15-07051-f005:**
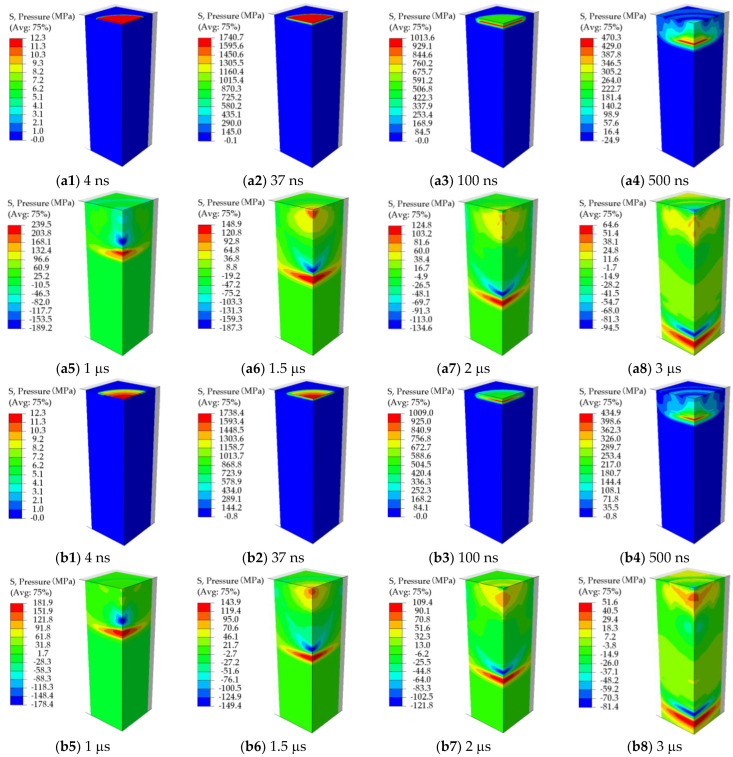
Propagation and attenuation of the LSP-induced shock wave: (**a1**–**a8**) Uniform distribution of laser plasma-induced shock wave pressure in the case of $P_{\max}=2.0\;\mathrm{GPa}\;\&\;D=8.0\;\mathrm{mm}$; (**b1**–**b8**) Gaussian distribution of laser plasma-induced shock wave pressure in the case of $P_{\max}=2.0\;\mathrm{GPa}\;\&\;D=8.0\;\mathrm{mm}$; (**c1**–**c8**) Uniform distribution of laser plasma-induced shock wave pressure in the case of $P_{\max}=0.6\;\mathrm{GPa}\;\&\;D=0.8\;\mathrm{mm}$; (**d1**–**d8**) Gaussian distribution of laser plasma-induced shock wave pressure in the case of $P_{\max}=0.6\;\mathrm{GPa}\;\&\;D=0.8\;\mathrm{mm}$.

**Figure 6 materials-15-07051-f006:**
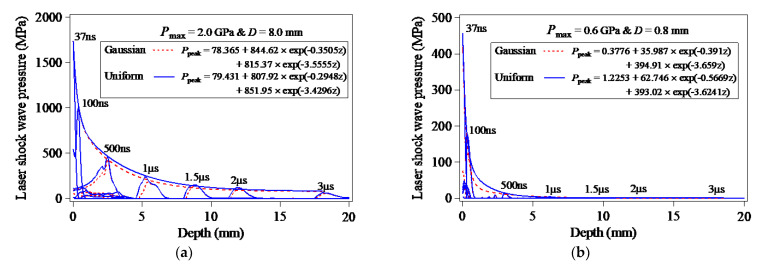
Evolution of the peak values of the LSP-induced shock wave pressure: (**a**) $P_{\max}=2.0\;\mathrm{GPa}\;\&\;D=8.0\;\mathrm{mm}$; (**b**) $P_{\max}=0.6\;\mathrm{GPa}\;\&\;D=0.8\;\mathrm{mm}$.

**Figure 7 materials-15-07051-f007:**
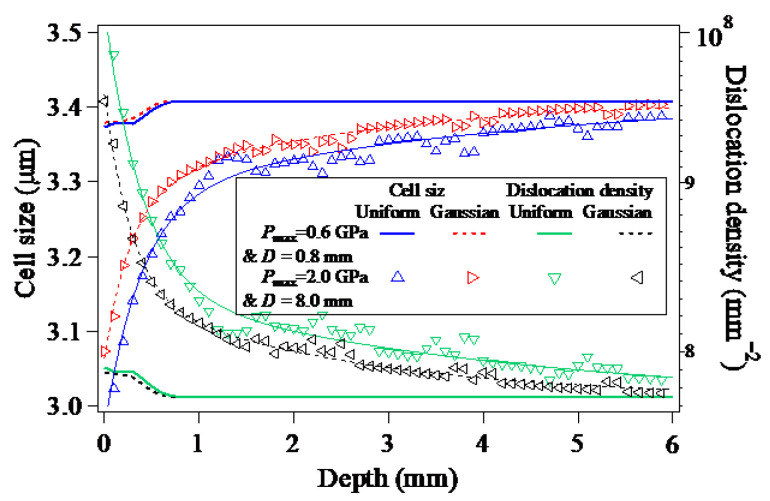
Evolution of the in-depth cell sizes and dislocation densities of target material during the LSP process.

**Figure 8 materials-15-07051-f008:**
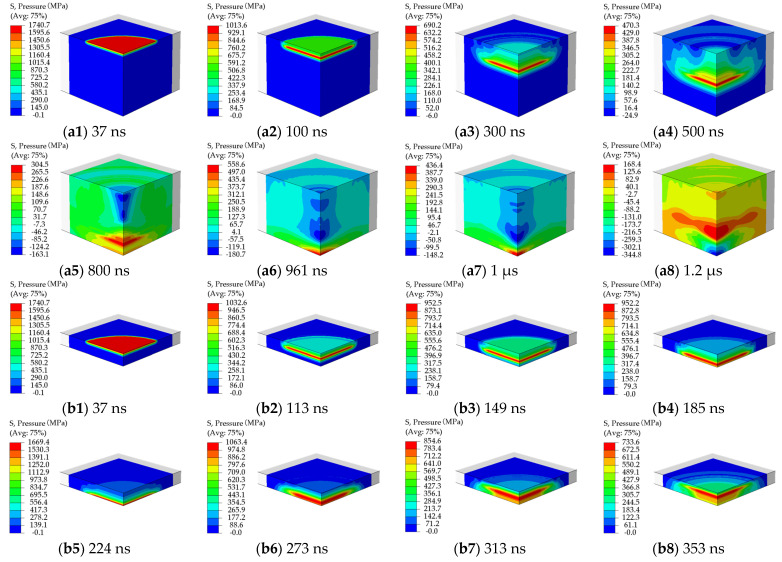
LSP-induced shock waves propagating in the target models: (**a1**–**a8**) 5 mm-thickness target model; (**b1**–**b8**) 1 mm-thickness target model.

**Figure 9 materials-15-07051-f009:**
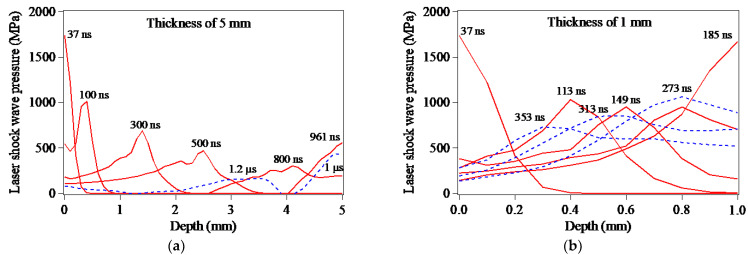
LSP-induced shock waves propagating along Path-Z in the target models: (**a**) 5 mm-thickness target model; (**b**) 1 mm-thickness target model.

**Figure 10 materials-15-07051-f010:**
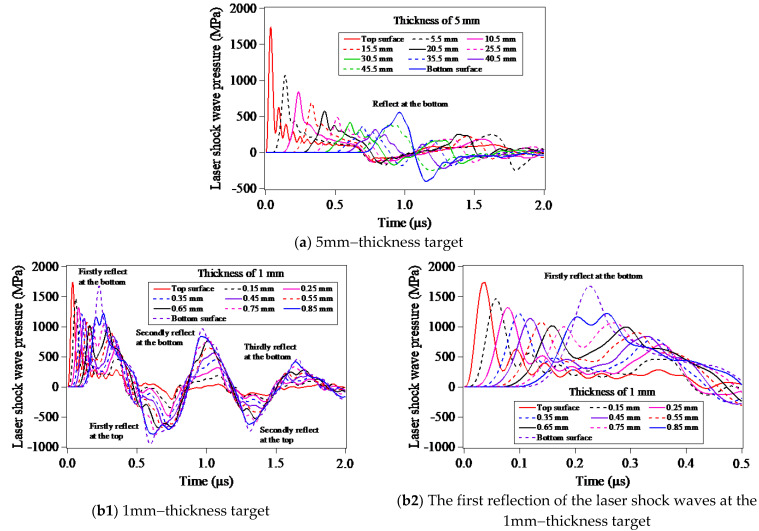
Propagation, attenuation and reflection of the LSP-induced shock waves in the target models: (**a**) 5 mm-thickness target model; (**b1**,**b2**) 1 mm-thickness target model.

**Figure 11 materials-15-07051-f011:**
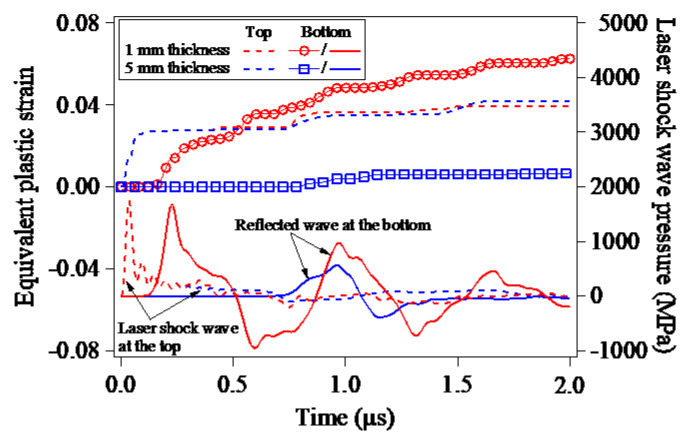
Evolutions of the equivalent plastic strains and the LSP-induced shock waves at the top and bottom surfaces of the target during the LSP process.

**Figure 12 materials-15-07051-f012:**
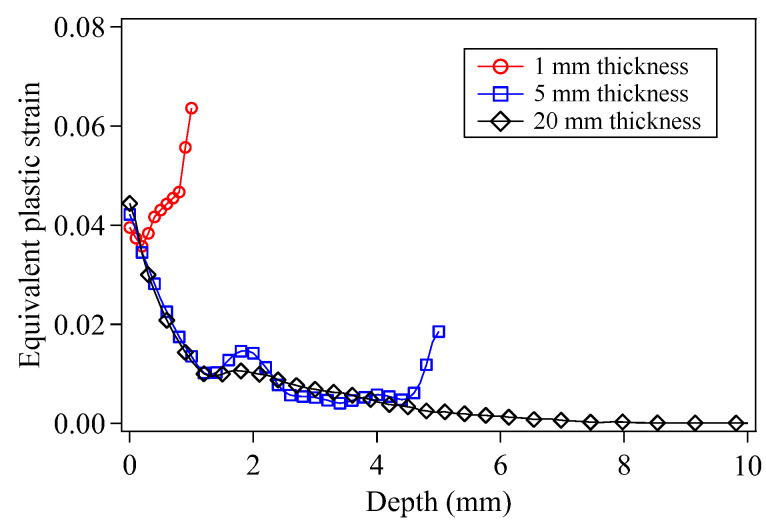
In-depth equivalent plastic strains of the LSPed targets with different thicknesses.

**Figure 13 materials-15-07051-f013:**
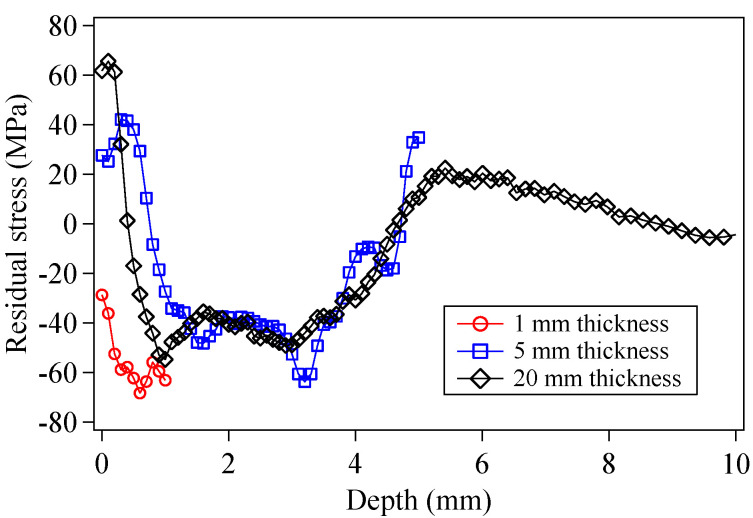
In-depth residual stresses of the LSPed targets with the different thicknesses.

**Figure 14 materials-15-07051-f014:**
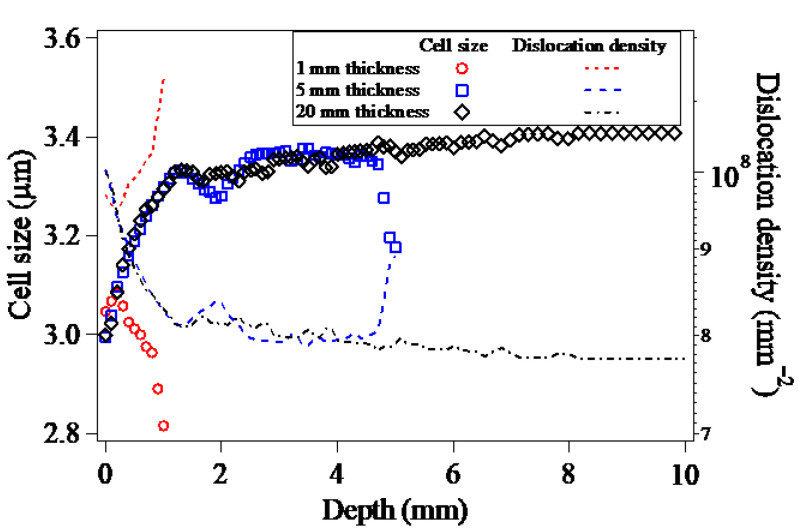
In-depth dislocation densities and cell sizes of the targets with different thicknesses.

**Table 1 materials-15-07051-t001:** Material parameters of the dislocation density-based constitutive model for pure Al [32].

$\alpha^{*}$	$\beta^{*}$	$k^{*}$	$m^{*}$	$n^{*}$	$\begin{array}{l} \sigma_{1}\\ \left(\mathrm{MPa}\right) \end{array}$	*G* (GPa)	$\begin{array}{l} {\dot{\gamma }}_{0}\\ \left(1/s\right) \end{array}$	$f_{0}$	$f_{\infty }$	$\begin{array}{l} \rho_{w0}\\ \left({\mathrm{mm}}^{-2}\right) \end{array}$	$\begin{array}{l} \rho_{c0}\\ \left({\mathrm{mm}}^{-2}\right) \end{array}$
0.0024	0.0054	3.22	100	67	55	26.3	1	0.25	0.06	$1\times 10^{7}$	$1\times 10^{8}$

## Data Availability

Data sharing is not applicable to this article.

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
