# Peer review of "Numerical Study on Laser Shock Peening of Pure Al Correlating with Laser Shock Wave"

_materials, 2022, doi:10.3390/ma15207051_

Round 1

Reviewer 1 Report

The ABSTRACT section is well–structured, is informative, can stand alone and covers the content. It briefly summarizes the studied problem. Also, the KEYWORDS are well–defined. Overall, this section is technically and fairly detailed.

The aims and objectives of the research are well defined. Also, the topics (laser shock processing, Laser shock peening (LSP), in comparison with the conventional shot peening (SP) and ultrasonic shot peening (USP), finite element modeling of LSP, dislocation density-based constitutive model) are important subjects, due to the main advantages of LSP (generation of deep compressive stresses and ability to target specific often hard-to-reach areas and complex geometries). Laser shock peening is a very useful surface treatment technique in practical applications, due to the compressive residual stress introduced by LSP which can significantly improve the mechanical performance of components, such as resistance to crack initiation and growth with extended fatigue life and enhanced fatigue strength. From the point of view of the importance of the theme, this work fits into the fundamental research, developed rapidly in this area in recent years.

The paper is structured properly and have the basic structure of a typical research paper (INTRODUCTION, MATERIAL & METHODS, RESULTS & DISCUSSIONS, CONCLUSIONS, REFERENCES, etc.). Overall, this manuscript is well–written and interesting to read. In this sense, the following comments are my argue:

¾    Clarity of manuscript is high: the terms are clearly defined and any nonrelevant information are avoided in the paper’s sections.

¾    Simplicity: The paper is well–structured and its parts are logically interconnected.

¾    Accuracy: all data and illustrations are clearly defined & presented.

The INTRODUCTION section provide the necessary background information needed to understand the paper. All background information identifies and describes the nature of a well-defined research problem (laser shock processing) with reference to the existing literature and placed the present research problem in a proper scientific context. Literature review provides comprehensive information about the current state of research.
The METHODOLOGY / MATERIALS & METHODS section is relatively well described and conducted (finite element modeling of LSP, based on dislocation density-based constitutive model and the dynamic flow stress in the constitutive model and Laser shock wave pressure model).
Because of the complexity of shock wave propagation in an alloy component, it is essential that the simulation can be correctly performed using a suitable computing capacity. The computer technology has developed rapidly in recent years, for many, this technology being a less exploited and known. In this sense, the authors make a good description of the procedures. Overall, this section is technically and fairly detailed.
The body of paper describe the important RESULTS of the research (Finite element simulation results, in Chapter 3, Effects of target thicknesses, in Chapter 4), followed by several DISCUSSIONS. The authors do a very good job of presenting their results and demonstrate the suitability of the method.  Overall, this section is technically and fairly detailed.

The authors proceed directly to CONCLUSIONS (Chapter 5), which succinctly summarize the major points of the paper, derived from the results. The authors fairly concludes in just a few sentences given the rich discussion in the body of the paper.

The list of REFERENCES is long and relatively well chosen. The entire bibliography is current (the oldest being from 1996), but modern works (over the last 5 years) are mainly used.

Figures are particularly important because they show the most objective support of the research. The graphic addenda is remarkable.

Author Response

The ABSTRACT section is well–structured, is informative, can stand alone and covers the content. It briefly summarizes the studied problem. Also, the KEYWORDS are well–defined. Overall, this section is technically and fairly detailed. 

The aims and objectives of the research are well defined. Also, the topics (laser shock processing, Laser shock peening (LSP), in comparison with the conventional shot peening (SP) and ultrasonic shot peening (USP), finite element modeling of LSP, dislocation density-based constitutive model) are important subjects, due to the main advantages of LSP (generation of deep compressive stresses and ability to target specific often hard-to-reach areas and complex geometries). Laser shock peening is a very useful surface treatment technique in practical applications, due to the compressive residual stress introduced by LSP which can significantly improve the mechanical performance of components, such as resistance to crack initiation and growth with extended fatigue life and enhanced fatigue strength. From the point of view of the importance of the theme, this work fits into the fundamental research, developed rapidly in this area in recent years. 

The paper is structured properly and has the basic structure of a typical research paper (INTRODUCTION, MATERIAL & METHODS, RESULTS & DISCUSSIONS, CONCLUSIONS, REFERENCES, etc.). Overall, this manuscript is well–written and interesting to read. In this sense, the following comments are my argue:

Clarity of manuscript is high: the terms are clearly defined and any non-relevant information is avoided in the paper’s sections.  

Simplicity: the paper is well–structured and its parts are logically interconnected. 

Accuracy: all data and illustrations are clearly defined & presented.  

The INTRODUCTION section provides the necessary background information needed to understand the paper. All background information identifies and describes the nature of a well-defined research problem (laser shock processing) with reference to the existing literature and placed the present research problem in a proper scientific context. Literature review provides comprehensive information about the current state of research. 

The METHODOLOGY / MATERIALS & METHODS section is relatively well described and conducted (finite element modeling of LSP, based on dislocation density-based constitutive model and the dynamic flow stress in the constitutive model and Laser shock wave pressure model). Because of the complexity of shock wave propagation in an alloy component, it is essential that the simulation can be correctly performed using a suitable computing capacity. The computer technology has developed rapidly in recent years, for many, this technology being a less exploited and known. In this sense, the authors make a good description of the procedures. Overall, this section is technically and fairly detailed. The body of paper describe the important RESULTS of the research (Finite element simulation results, in Chapter 3, Effects of target thicknesses, in Chapter 4), followed by several DISCUSSIONS. The authors do a very good job of presenting their results and demonstrate the suitability of the method. Overall, this section is technically and fairly detailed. 

The authors proceed directly to CONCLUSIONS (Chapter 5), which succinctly summarize the major points of the paper, derived from the results. The authors fairly conclude in just a few sentences given the rich discussion in the body of the paper.

The list of REFERENCES is long and relatively well chosen. The entire bibliography is current (the oldest being from 1996), but modern works (over the last 5 years) are mainly used.

Figures are particularly important because they show the most objective support of the research. The graphic addenda is remarkable. 

RESPONSE:  Thanks for your positive evaluation.

Reviewer 2 Report

The experimental test results from literature are mechanically sound. However, the numerical simulation results are in opposite sign at shallow z for second Pmax case. This shows the numerical model is not correct and the later extensive simulation results are in doubt. The reviewer advises the authors to revise their model to achieve a good agreement with the experimental test results before subsequent more simulations and resubmission.

Author Response

The experimental test results from literature are mechanically sound. However, the numerical simulation results are in opposite sign at shallow z for second Pmax case. This shows the numerical model is not correct and the later extensive simulation results are in doubt. The reviewer advises the authors to revise their model to achieve a good agreement with the experimental test results before subsequent more simulations and resubmission.  

RESPONSE:  Thanks for your comments.  The finite element modeling method for the simulation of LSP is consistent with some researches that have published in the well-known journals, such as Applied Surface Science [2014, 316: 477-483], Optics & Laser Technology [2021, 142: 107217; 2019, 111: 146-155] and Journal of Materials Processing Technology [2015, 223: 8-15]. 

By using the finite element model of LSP, in the case of Pmax=2.0GPa & D=8.0mm, the predicted in-depth residual stresses with the distance smaller than 1mm from the LSPed surface are different from the experimental results, as shown in Figure 3, which could be related to the material constitutive model and the experiment measurement method.  The pure Al material parameters of the dislocation density-based constitutive model are taken from the literature [Mater. Sci. Eng. A. 2003, 351, 86-97.], which may be not very suitable for the experiment study of LSP of pure Al taken from the literature [Appl. Surf. Sci. 2014, 316, 477-483].  The predicted residual stress is outputted from an element distributed along the Path-Z, whereas the experimentally measured residual stresses are usually the averaged value within a small area. 

On the other hand, in the case of Pmax=2.0GPa & D=8.0mm, it is noted that the predicted in-depth residual stresses with the distance larger than 1mm from the LSPed surface are in perfect agreement with the experimental results, and the predicted depth of the compressive residual stress induced by LSP also agrees well with the experimental data.  Additionally, as seen from Figure 3, it is observed that the finite element prediction of the in-depth residual stresses induced by LSP are in good consistence with the experiment results in the case of Pmax=0.6GPa & D=0.8mm, which could verify the validation of the finite element model of LSP.  

Considering that this paper is devoted to investigating the effects of the dynamic propagation, attenuation and reflection of the LSP-induced shock wave on the resultant plastic deformation, compressive residual stress field, dislocation density evolution and grain refinement, therefore the finite element model of LSP proposed in this work should be considered to be feasible and accepted.

Reviewer 3 Report

This study reports a series of hydrodynamic simulations of laser-induced shockwave propagation under different conditions. The approach remains within limits of established methodology and nothing really new is reported. However, it is useful for publish the present results, because they provide a good reference for those who may not have the resources of conducting such simulations themselves.

In terms of what may be achievable for the authors within the frame of their approach I would only ask them to consider the anisotropy of strain and the elastic tensor explicitly rather than their present (somewhat cheap) isotropic approach - but this is up to you!

Hence, I support publication of this paper. However, the wording needs extensive correction - the paper really cannot be published in the present poor writing. I list incorrect grammar and wording for the Introduction only - I cannot correct the whole paper. Please get some help in improving the writing!

Detailed comments:

The laser plasma-induced shock wave is responsible for the results of laser shock peen-11ing (LSP) in terms of the plastic deformation, compressive residual stress field and microstructure 12evolution. 

-> The laser plasma-induced shock wave CAUSES laser shock peen-
ing (LSP) THROUGH plastic deformation, compressive residual stress field and microstructur-AL evolution. 

 Reducing 21the target thickness can significantly increase the reflection times …

-> You mean rarefaction or reverberation? Please clarify!

promising surface enhancement tech-29niques,

-> promising TECHNIQUES FOR surface enhancement

resistances of the  wear, corrosion  and fa-30tigue
-> CHANGE WORDING to proper English.

the ablative layer covering on the material surface to be peening. 
-> the ablative layer covering on the material surface to be peen-ED

makes the ablative layer to be vaporized and ionized 40into the plasma.
-> vaporizes and ionizes the ablative…

then  develops  to  the  shock  wave.
-> wrong wording!

  when  the  laser  shock  wave  pressure  is 44larger  than  the  dynamic  yield  strength  of  the  metallic  material,  the  plastic  deformation wave…

->  please correct the wording!

wave  can  be  detected  by  the  several  devices  in  the 53experimental  investigation,  s
-> wave  can  be EXPERIMENTALLY detected  by  the  several  devices.

Streak-camera  photograph,  interferometer  systems, 54optical  shadow  graphy, 

-> ->  please correct the wording!
-> Also Doppler shift interferometry.

However, it is hardly to carry out

-> >  please correct the wording!

within the metallic materials

> >  please correct the wording!

Actually, in situ Laue X-ray diffraction gives quantitative measures of strain and stress(given a known elastic tensor) whether for metals or optically transparent materials. See Zhang et al. DOI10.1103/PhysRevLett.127.045702.

by  the  laser-induced  plasma 66pressure,and the prediction results were verified by the PVDF measurements of the shock 67force  at  the  rear  face  of  the  plates  with  different  thicknesses.
-> reformulate!

 residual stress-strain profile is explained that the high strength stress is 73formed when the laser shock wave is reflected and coupled in the metal sheet, causing the 74re-plastic deformation and the decrease of transverse plastic strain.

-> reformulate!

Finite element modeling of LSP

-> this is standard work - it is fine. However, you do not account for the anisotropy of strain and of elastic properties by using scalar rather than tensorial parameters like G and epsilon.
However, dynamic stress can build up at grain contacts to locally very high values which are relevant over the short time scale of the laser driven shock state.
If you cannot account for the actual anisotropy of the material response on grain level, at least you should address this issue in your paper.

I also like to mention that in real materials voids are important: Upon dynamic compression they generate hot spots which cause a propagation of reduced impedance and higher temperature.
You should also address this at least by comment, if you cannot implement it in your model.

Uniform distribution of laser plasma-induced shock wave pressure in the case of GPa0.2max=P&mm0.8=

-> The uniform distribution is result of your uniform input parameters and of neglecting any boundary effects. If you take into consideration the finite boundaries and shock release + plastic-elastic deformation at the boundaries, this may be different. Of course, your basic result that planar shock compression is sustained the better the larger the diameter of the laser spot (equivalent to a flyer) is true but this is no new result.

  reflected  wave  pressure

-> you mean reverberation?!

Please use established terminology for dynamic compression phenomena instead of imprecise neologisms:
Obviously, there is no strict ‘reflection’ of shockwaves, there is reverberation or rarefaction.

Author Response

Reviewer #3 

This study reports a series of hydrodynamic simulations of laser-induced shockwave propagation under different conditions.  The approach remains within limits of established methodology and nothing really new is reported.  However, it is useful for publish the present results, because they provide a good reference for those who may not have the resources of conducting such simulations themselves. 

RESPONSE: Thanks for your positive evaluation. 

In terms of what may be achievable for the authors within the frame of their approach I would only ask them to consider the anisotropy of strain and the elastic tensor explicitly rather than their present (somewhat cheap) isotropic approach - but this is up to you!

RESPONSE:  Thanks for your useful suggestions.  This paper aims at developing the three-dimensional finite element model in conjunction with the dislocation density-based constitutive model for simulating the LSP of pure Al correlating with the laser shock wave.  For simplicity, the isotropic approach was used in the present work.  Our further works would take into account the suggestions that the anisotropy of strain and the elastic tensor explicitly. 

Hence, I support publication of this paper. However, the wording needs extensive correction - the paper really cannot be published in the present poor writing. I list incorrect grammar and wording for the Introduction only - I cannot correct the whole paper. Please get some help in improving the writing! 

RESPONSE: Thanks for your useful suggestions. All comments have been addressed to improve the manuscript for publication in Materials

Detailed comments:

  1. The laser plasma-induced shock wave is responsible for the results of laser shock peening (LSP) in terms of the plastic deformation, compressive residual stress field and microstructure evolution.

> The laser plasma-induced shock wave causes laser shock peening (LSP) through plastic deformation, compressive residual stress field and microstructural evolution. 

RESPONSE:  Thanks for your comments. In order to improve the readability of manuscript, this sentence in the original manuscript has been rewritten as

"The laser shock peening (LSP) is an innovative and promising surface strengthening technology of metallic materials, the LSP-induced plastic deformation, compressive residual stresses and microstructure evolution are essentially attributed to the laser plasma-induced shock wave."

  1. Reducing the target thickness can significantly increase the reflection times.

> You mean rarefaction or reverberation? Please clarify!  

RESPONSE:  Thanks for your comments, and the reflection of the LSP-induced stress wave was investigated in this paper. 

  1. promising surface enhancement techniques,

> promising techniques for surface enhancement

RESPONSE:  Thanks for your comments, the "promising surface enhancement techniques" has been rewritten as "promising techniques for surface enhancement" in the revised manuscript. 

4 resistances of the wear, corrosion and fatigue

> Change wording to proper English.

RESPONSE:  Thanks for your comments, the words have been rewritten as "wearing quality, corrosion resistance and fatigue performance" in the revised manuscript. 

  1. the ablative layer covering on the material surface to be peening.

> the ablative layer covering on the material surface to be peened.

RESPONSE:  Thanks for your comments, the "to be peening" has been revised as "to be peened". 

  1. makes the ablative layer to be vaporized and ionized into the plasma.

> vaporizes and ionizes the ablative

RESPONSE: Thanks for your comments, and the sentence has been modified in the revised manuscript. 

"The excessive heat induced by the laser irradiation vaporizes and ionizes the ablative layer that then turns to the plasma."

  1. then develops to the shock wave.

> wrong wording!

RESPONSE: Thanks for your comments, and the sentence has been modified in the revised manuscript. 

"The plasma expands rapidly by continuing to absorb the laser radiation, and the plasma-induced shock wave with the ultra-high peak pressure is resultantly formed."

  1. when the laser shock wave pressure is larger than the dynamic yield strength of the metallic material, the plastic deformation wave

> please correct the wording!

RESPONSE: Thanks for your comments, and the sentence has been modified in the revised manuscript. 

"when the peak pressure of the LSP-induced shock wave (stress wave) exceeds the dynamic yield strength of the metallic materials, the plastic deformation with ultra-high strain rate occurs."

  1. wave can be detected by the several devices in the experimental investigation,

> wave can be experimentally detected by the several devices.

RESPONSE: Thanks for your comments, and the sentence has been modified in the revised manuscript. 

"the LSP-induced shock wave can be experimentally detected by several devices."

  1. Streak-camera photograph, interferometer systems, optical shadow graphy,

> please correct the wording!

> Also Doppler shift interferometry.

RESPONSE: Thanks for your comments, and the sentence has been modified in the revised manuscript.

"a velocimetry interferometer system for any reflector (VISAR) interferometer device and the piezoelectric thin-film polyvinylidene fluoride (PVDF)."

  1. However, it is hardly to carry out

> please correct the wording!

 within the metallic materials

> please correct the wording!

Actually, in situ Laue X-ray diffraction gives quantitative measures of strain and stress (given a known elastic tensor) whether for metals or optically transparent materials. See Zhang et al. DOI10.1103/PhysRevLett.127.045702.  

RESPONSE:  Thanks for your comments, and the sentence has been modified in the revised manuscript. 

"Although it is feasible to carry out the in situ exploration of the propagation and attenuation of the LSP-induced shock wave in the process of LSP of metals by utilizing the available experiment devices [Zhang et al. DOI:10.1103/PhysRevLett.127.045702.], whereas the finite element method has been increasingly favored and frequently used by the researchers with the development of computer technology and numerical calculation [17-20].

  1. by the laser-induced plasma pressure, and the prediction results were verified by the PVDF measurements of the shock force at the rear face of the plates with different thicknesses.

> reformulate!

RESPONSE: Thanks for your comments, and the sentence has been modified in the revised manuscript. 

"and the prediction results were experimentally verified by the PVDF measurements."

  1. residual stress-strain profile is explained that the high strength stress is formed when the laser shock wave is reflected and coupled in the metal sheet, causing the re-plastic deformation and the decrease of transverse plastic strain.

> reformulate!

RESPONSE: Thanks for your comments, and the sentence has been rewritten in the revised manuscript. 

"Compared with the infinite thickness model, the propagation law of LSP-induced shock wave and the dynamic response of material were analyzed, and the simulation results show that the reflection of the LSP-induced shock wave has an important impact on the LSP-induced residual stresses in the thin (sheet) model." 

  1. Finite element modeling of LSP

> this is standard work - it is fine. However, you do not account for the anisotropy of strain and of elastic properties by using scalar rather than tensorial parameters like G and epsilon.  However, dynamic stress can build up at grain contacts to locally very high values which are relevant over the short time scale of the laser driven shock state. If you cannot account for the actual anisotropy of the material response on grain level, at least you should address this issue in your paper.

RESPONSE: Thanks for your comments, and the explanation on this issue has been added in the revised manuscript. 

"For simplicity, it should be noted that the finite element modeling of LSP in this work has not taken into account the influences of the actual anisotropy of the material grain-level response on the numerical simulations." 

  1. I also like to mention that in real materials voids are important: Upon dynamic compression they generate hot spots which cause a propagation of reduced impedance and higher temperature. You should also address this at least by comment, if you cannot implement it in your model.

RESPONSE: Thanks for your comments, and the explanation on this issue has been added in the revised manuscript. 

"It is noted that the dislocation density-based constitutive model was proposed with the assumptions that the metallic materials are continuous, homogeneous and isotropic.  The micro voids in the real metallic materials are not taken into consideration, which would have an impact on the finite element computation results."

  1. Uniform distribution of laser plasma-induced shock wave pressure in the case of Pmax=2.0 & D=8.0mm.

> The uniform distribution is result of your uniform input parameters and of neglecting any boundary effects.  If you take into consideration the finite boundaries and shock release + plastic-elastic deformation at the boundaries, this may be different. Of course, your basic result that planar shock compression is sustained the better the larger the diameter of the laser spot (equivalent to a flyer) is true but this is no new result.

RESPONSE: Thanks for your comments, and the explanation on this issue has been added in the revised manuscript. 

"whereas the laser shock wave pressure is considered to be distributed uniformly within the scope of the larger laser spot neglecting any boundary effects."

  1. reflected wave pressure

> you mean reverberation?!

Please use established terminology for dynamic compression phenomena instead of imprecise neologisms: Obviously, there is no strict ‘reflection’ of shockwaves, there is reverberation or rarefaction. 

RESPONSE:  Under the laser plasma-induced shock wave pressure, the LSP-induce stress wave propagates and attenuates along the thickness direction.  After the LSP-induced stress wave arrives the bottom surface of target, the reflection of stress wave occurs, resulting in the twice pressure of the stress wave at the bottom surface.  The reflected stress wave then propagates toward the top surface from the bottom surface of the target, as shown in the following figure. 

(Optics & Laser Technology, 2021, 142: 107217)
